# Automated identification and tracking of cells in Cytometry of Reaction Rate Constant (CRRC)

**Giammarco Nebbioso**[1,2◦], **Robel Yosief**[1,2◦], **Vasilij Koshkin**[1,2], **Yumin Qiu**[2,3], **Chun Peng**[2,3], **Vadim Elisseev**[4,5], **Sergey N. Krylov**[1,2]*

**1** Department of Chemistry, York University, Toronto, Ontario, Canada, **2** Centre for Research on Biomolecular Interactions, York University, Toronto, Ontario, Canada, **3** Department of Biology, York University, Toronto, Ontario, Canada, **4** IBM Research Europe, The Hartree Centre, Daresbury Laboratory, Warrington, United Kingdom, **5** Wrexham Glyndwr University, Wrexham, United Kingdom

◦ These authors contributed equally to this work.
* skrylov@yorku.ca

**Data Availability Statement:** All relevant files are available from the FigShare database (https:// figshare.com/articles/media/Workflow_for_ Cytometry_of_Reaction_Rate_Constant_CRRC_

## Abstract

Cytometry of Reaction Rate Constant (CRRC) is a method for studying cell-population heterogeneity using time-lapse fluorescence microscopy, which allows one to follow reaction kinetics in individual cells. The current and only CRRC workflow utilizes a single fluorescence image to manually identify cell contours which are then used to determine fluorescence intensity of individual cells in the entire time-stack of images. This workflow is only reliable if cells maintain their positions during the time-lapse measurements. If the cells move, the original cell contours become unsuitable for evaluating intracellular fluorescence and the CRRC experiment will be inaccurate. The requirement of invariant cell positions during a prolonged imaging is impossible to satisfy for motile cells. Here we report a CRRC workflow developed to be applicable to motile cells. The new workflow combines fluorescence microscopy with transmitted-light microscopy and utilizes a new automated tool for cell identification and tracking. A transmitted-light image is taken right before every fluorescence image to determine cell contours, and cell contours are tracked through the timestack of transmitted-light images to account for cell movement. Each unique contour is used to determine fluorescence intensity of cells in the associated fluorescence image. Next, time dependencies of the intracellular fluorescence intensities are used to determine each cell's rate constant and construct a kinetic histogram "number of cells vs rate constant." The new workflow's robustness to cell movement was confirmed experimentally by conducting a CRRC study of cross-membrane transport in motile cells. The new workflow makes CRRC applicable to a wide range of cell types and eliminates the influence of cell motility on the accuracy of results. Additionally, the workflow could potentially monitor kinetics of varying biological processes at the single-cell level for sizable cell populations. Although our workflow was designed ad hoc for CRRC, this cell-segmentation/cell-tracking strategy also represents an entry-level, user-friendly option for a variety of biological assays (i.e., migration, proliferation assays, etc.). Importantly, no prior knowledge of informatics (i.e., training a model for deep learning) is required.

**Funding:** This work was supported by the Natural Sciences and Engineering Research Council of Canada https://www.nserc-crsng.gc.ca/index_eng. asp (grant STPG-P 521331-2018 to SKN; Sergey N. Krylov) and the Canadian Institutes of Health Research https://cihr-irsc.gc.ca/e/193.html (grant PJT-166079 to CP; Chun Peng). The funders had no role in study design, data collection and analysis, decision to publish, or preparation of the manuscript.

**Competing interests:** The authors have declared that no competing interests exist.

## 1. Introduction

Cancerous tissues are typically very heterogeneous; a single tumor may be composed of several distinct cell populations, for example, a population of bulk tumor cells and a population of tumor-initiating cells [1, 2]. Quantitative characteristics of tumor composition, e.g., the size of the population of tumor-initiating cells, define its carcinogenic features, e.g., resistance to chemotherapy [3, 4]. Fundamentally, tumor heterogeneity is caused by differences in molecular reactions between the cells. If a reaction is associated with tumor heterogeneity, it can serve as a basis for characterizing this heterogeneity [5].

Cytometry is a general approach to study tumor heterogeneity by measuring fluorescence at the single-cell level. Cytometry of Reaction Rate Constant (CRRC) is a technique that follows reaction kinetics at the single-cell level and presents the results as a kinetic histogram "number of cells versus rate constant" [6–11]. Rate constants are the most robust parameters to characterize chemical reactions, and, accordingly, CRRC can support robust and accurate characterization of reaction-based cell-population heterogeneity [12]. CRRC may be potentially suitable for the development of reliable cancer biomarkers built upon such heterogeneity [13].

CRRC is based on time-lapse fluorescence microscopy (Fig 1). Conceptually, a fluorescent or fluorogenic substrate, which is involved in the reaction of interest, is loaded into the cells. Fluorescence images of a few hundred cells are taken progressively to monitor the change in intracellular fluorescence intensity. The images are processed to obtain a kinetic trace "fluorescence intensity versus time" for each cell, which is used to determine the rate constant for each cell. Finally, the data are presented as a kinetic histogram: "number of cells versus rate constant."

CRRC is still in its infancy. The current and only CRRC workflow, which was used for proving CRRC in-principle, includes confocal fluorescence microscopy, and utilizes a single fluorescence image to manually identify cell contours [12]. The cell contours identified from this single image are used to determine fluorescence intensity of individual cells in every other image of the large time-stack of images. This rudimentary workflow assumes that each cell retains its position in the image throughout the entire course of time-lapse measurements [12]. Such an assumption is impossible to satisfy for motile cells which move significantly during the time-lapse measurements. Intracellular fluorescence intensity will become inaccurate as cells gradually deviate from the cell contours used to determine fluorescence intensity. Thus, making CRRC robust to cell movement requires a new workflow that identifies cell contours for each fluorescence image and tracks cell contours through the time-stack of images.

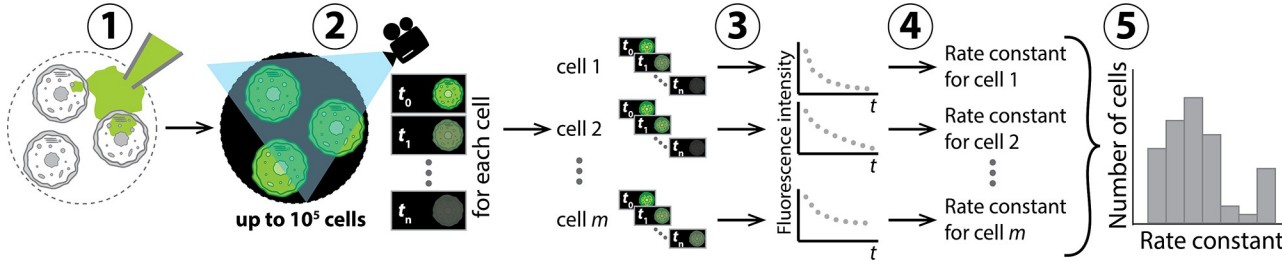

**Fig 1.** Schematic representation of five major steps in the CRRC analysis: 1) a fluorescent substrate involved in the reaction of interest is loaded into the cells, 2) a time-lapse microscopy experiment is initiated, and sequential images are captured to monitor the change in intracellular fluorescence intensity, 3) intracellular fluorescence intensity is calculated for each single cell as a function of time, 4) rate constants (*k*) are determined from reaction kinetics, i.e., dependencies of fluorescence intensity on time, and 5) a kinetic histogram "number of cells versus rate constant" is plotted to facilitate accurate analysis of tissue heterogeneity.

Several biological assays, such as migration, proliferation, and cell-cycle assays, are based on single-cell time-lapse microscopy [14–17]. The reliability of such assays largely depends on the assay's ability to properly track each single cell over a stack of images. To serve this purpose, different tracking tools, which rely on automatic single-cell segmentation, have been developed [16, 17]. It is noteworthy that most of these tracking tools are designed to track fluorescently labelled objects [18]. However, certain applications (e.g., CRRC) require cell tracking to be performed on a set of non-fluorescence (unstained) images (e.g., bright-field (BF), differential interference contrast (DIC), and phase-contrast (PC) microscopy). In this case, all tools designed to track fluorescently labelled objects are expected to fail. To overcome this issue, advanced tracking tools based on deep learning have been proposed [19–23]. Although they represent a valid solution for cell tracking of images of unstained cells, they are far from being user-friendly as they require the user to have a high level of expertise in informatics. Moreover, training a neural network requires a considerable amount of time. Therefore, both the complexity and time required to train a deep learning network can present an obstacle for many users. For example, after completing our manuscript, we found a recent publication reporting the development of an automated cell-tracking tool reminiscent of the one proposed in our work but requiring a model to be trained [20]. To the best of our knowledge, there is no workflow that allows tracking single cells through a stack of unstained images without relying on the complexity of neural networking. Here we report on the development of such a workflow.

The new CRRC workflow combines two types of optical microscopy: (*i*) transmitted-light microscopy for cell-contour identification and cell tracking through the time-stack of images and (*ii*) fluorescence microscopy for monitoring substrate conversion into the product during the time-lapse imaging. Imaging is done in an automated fashion with a transmitted-light image taken right before every fluorescence image. Time-correlated stacks of transmitted-light and fluorescence images are processed and analyzed automatically to produce kinetic traces "fluorescence intensity versus time" which are unaffected by cell displacement.

Workflow development and validation included three major steps. First, we optimized the use of transmitted-light microscopy for cell-contour identification. Second, we proved that cell displacement between the adjacent transmitted-light and fluorescence images is negligible even for highly-motile cells; hence, cell contours determined from transmitted-light images are applicable to fluorescence images. Finally, we conducted a comparative study of the original and new workflows in CRRC of cross-membrane transport in motile cells. The results clearly demonstrated that limitations of the original CRRC workflow combined with those of kinetic-analysis algorithms led to a systematic shift of CRRC histograms to the right. These systematic errors in the original CRRC workflow may wrongly identify subpopulations of cells with very high rate constants. In contrast, the new CRRC workflow facilitates the determination of accurate kinetic histograms.

## 2. Materials and methods

### 2.1. Cell culture

Ovarian cancer cells TOV-112D were purchased from ATCC and maintained in MCDB 105/ Medium 199 (Sigma-Aldrich, St. Louis, MO, USA, Cat. No. of MCDB 105: M6395, Cat. No. of Medium 199: M5017) supplemented with 10% fetal bovine serum (Gibco, Grand Island, NY, USA, Cat. No: 12483–020). Cells were cultured in 60-mm (Sarstedt AG&Co, Numbrecht, Germany, Cat. No: 83.3901) and 35-mm dishes for imaging (Nest Biotechnology Co, Wuxi, Jiangsu, China, Cat. No: 706001) at 37˚C in a humidified incubator with 5% $CO_2$. Cells were cultured until they reached approximately 70% confluence.

## 2.2. Cell staining

To perform nuclei staining for cell counting in the original workflow, 10 μL of 6.5 mM saponin (Sigma-Aldrich, St. Louis, MO, USA, Cat. No: 8047152) and 5 μL of 1 mM propidium iodide (PI, Sigma-Aldrich, St. Louis, MO, USA, Cat. No: 25535164) were added into the Hanks' Balanced Salt Solution (HBSS) (Gibco, Grand Island, NY, USA, Cat. No:14025092) after completion of the time-lapse experiment (see CRRC Experimental flow for more details) [24]. After 10 min, cells were imaged with no washing.

## 2.3. CRRC experimental flow

Cell imaging was conducted on 35-mm plastic-bottom dishes with one exception when a 50-mm glass-bottom dish was used instead (Mattek, Ashland, MA, USA, Cat. No: P50G-1.5-14-FGRD). Four steps were followed to prepare cells for a CRRC cross-membrane transport experiment. First, we removed culture medium and washed cells once with 1 mL of PBS. Second, we incubated cells for 30 min in 1.2 mL of HBSS containing 1.5 μM fluorescein (Sigma-Aldrich St. Louis, MO, USA, Cat. No: 518478), the substrate of cross-membrane transport, and 10 μM glibenclamide (Research Biochemicals International, Natick, MA, USA, Cat. No: G106), a cross-membrane transport inhibitor. Third, we removed HBSS, and washed cells three times with 1 mL of PBS each. Fourth, we added 1.2 mL of HBSS and started image acquisition with alternating transmitted-light and fluorescence modes every 1 min for 1 h.

## 2.4. Image acquisition

In the previous CRRC studies, imaging was performed with confocal laser-scanning fluorescence microscopy [12, 25, 26]. In the current work, we used epifluorescence microscopy with a Leica DMi8 high-throughput cell-imaging system. This imager allows carrying out fully automated time-lapse image acquisition with alternating transmitted-light and fluorescence microscopy. BF, DIC, and fluorescence images were acquired with the same apochromatic HC PL APO 10x/0.45 objective lens. PC images were acquired with a N Plan 10×/0.25 PH1 objective lens. A FITC filter cube was used for fluorescein and a RHOD cube for PI (a nuclei stain). All images were captured with a deep-cooled high-resolution sCMOS camera. See Note S1 in S1 File for details on microscope settings and microscopy protocol.

## 2.5. Image processing software

We chose Fiji [27], an open-source software, because it can be easily adopted by others and supports all image processing and image analysis required for a CRRC workflow: (*i*) merging transmitted-light and fluorescence images, (*ii*) cell segmentation, i.e., determination of cell contours and, thus, identification of cells using the StarDist detector, (*iii*) cell tracking, including creation of tracks and exclusion of cells with incomplete tracks, and (*iv*) integration of intracellular fluorescence within the cell contours. Advantageously, a recent version of the Fiji plugin named TrackMate integrates capabilities for steps (*ii*) – (*iv*), which greatly simplifies image processing and analysis. Software settings and other details can be found in Note S2 in S1 File.

## 2.6. Extraction and analysis of kinetic traces

Intracellular fluorescence intensities were extracted from TrackMate and arranged in Microsoft Excel to build individual kinetic traces. The kinetic traces were fitted with the exponential decay (ExpDec1) function in OriginPro® software from the time of medium exchange at the beginning of the experiment (initiation of cross-membrane transport). A custom-made fitting

program has been developed using SciPy open-source Python library [28], and was used to cross-validate results obtained with OriginPro. The best fits produced rate constants of substrate efflux, $k_{efflux}$, for individual cells. Negative values of $k_{efflux}$ and all $k_{efflux}$ values with high uncertainty (relative standard error, RSE > 100%) were removed from further analysis.

## 2.7. Cell population analysis

Cross-membrane transport of each cell population was characterized by frequency histograms of $k_{efflux}$ values of individual cells. Histograms were plotted in OriginPro software using the Custom Binning mode and were characterized by the median (peak position) and skewness (peak asymmetry) values obtained with the Descriptive Statistics tool. The comparison of distributions was conducted using the Kolmogorov-Smirnov test, considering $\alpha = 0.001$ as a criterion of statistical significance.

# 3. Results and discussion

## 3.1. Need for transmitted-light microscopy

The first key requirement for ensuring CRRC insensitivity to cell movement is that cell contours be identified in each fluorescence image in the time-stack of images. The very nature of CRRC prohibits the use of fluorescence from the substrate (product) to identify the cell contours. Since CRRC follows kinetics of fluorescence decrease (or increase), a portion of the fluorescence images in the time-stack always has too weak intracellular fluorescence for cell-contour identification. As such, we identify the cell contours in each fluorescence image with a standard multichannel imaging experiment and take an accompanying high-contrast image right before each fluorescence image of the substrate (product).

The accompanying image can be either a fluorescence one or a transmitted-light one, however, using an accompanying fluorescence image necessitates cells' pre-staining with a fluorescence probe spectrally different from the substrate (product). Such a probe would impose an additional chemical stress on the cells and could also interfere with measurements of substrate (product) fluorescence intensity due to unavoidable spectral overlaps. Therefore, our *a priori* preference was an accompanying transmitted-light image. Focal planes in fluorescence and transmitted-light modes may differ, but modern microscopes provide options of separate focusing in both fluorescence and transmitted-light modes.

Using transmitted-light images for cell-contour identification imposes a challenge: the contrast between cells and background in transmitted-light images is much lower than in fluorescence images. All software tools available for cell-contour identification perform best when cells appear as bright objects on a dark background. Standard transmitted-light images do not provide the required contrast independently on the imaging mode: DIC, PC, or BF. Yet, there is a relatively simple solution for this problem since image processing can increase the contrast of transmitted-light images. Increasing contrast leads to decreasing dynamic range of signal inside the cell image, but, advantageously, CRRC only needs cell contours from transmitted-light images in this workflow. Multiple algorithms exist for increasing cell image contrast [29–32]. We chose a method called thresholding for convenience: thresholding is a standard software tool in most advanced microscopes. It was shown that thresholding benefits from having a transmitted-light image slightly out of focus [31]. Having an image out of focus and subjected to thresholding raises a question of whether DIC and PC, which have better contrast in raw images than BF, would retain this advantage. Thus, we compared these three modes for their utility in cell-contour identification.

## 3.2. Preferred mode of transmitted-light microscopy

The three transmitted-light modes were assessed for their performance in correctly identifying cells compared to manual counting of cells contrasted with PI. PI is a bright fluorescent dye that stains nuclei in cells with a permeabilized plasma membrane. The nuclei in images of PI-stained cells are always spaced out by the cytoplasm; therefore, fluorescence images of PI-stained cells appear as well-separated bright spots in a mono-layer cell culture. Such images are well suited for manual cell counting (a cumbersome task) and for computer assisted cell counting [33]. An example of a raw fluorescence image of PI-stained cells is shown in the left-most panel of Fig 2A. The cells were counted manually in raw fluorescence images, and these numbers were used as a reference. BF, DIC, and PC images of the same fields of view were taken immediately after the fluorescence image but with a 30 μm lower focal plane. The cells appear out of focus, but they are brighter than the background which is beneficial for thresholding (see three rightmost panels in Fig 2A as and example).

All four raw images (fluorescence, BF, DIC, and PC) were processed before being subjected to automated cell-contour determination. The fluorescence images were simply converted from RGB to the 16-bit format (see the leftmost panel in Fig 2B as an example). Transmitted-light images were subjected to live-mode thresholding to obtain high-contrast images (see three rightmost panels in Fig 2B as an examples). We refer the reader to Note S1 in S1 File for

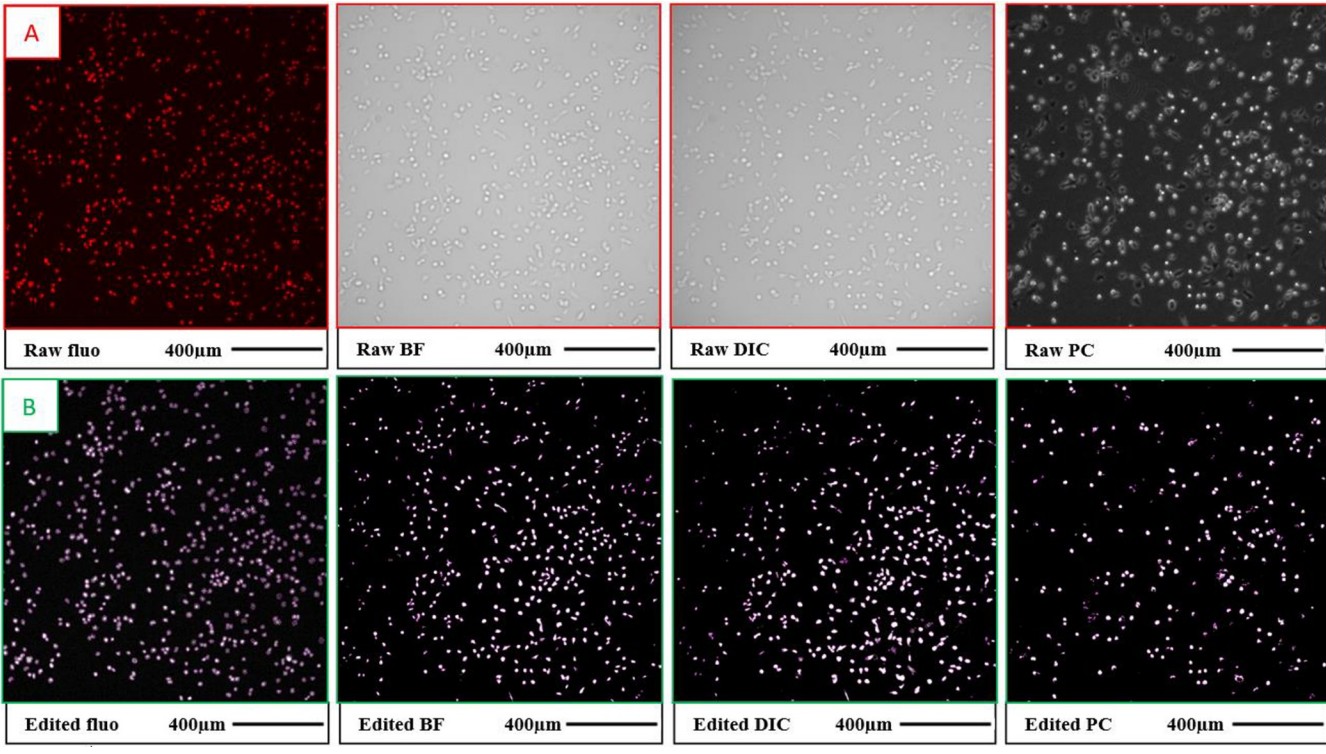

**Fig 2. Comparing three modes of transmitted-light microscopy (BF, DIC, and PC) for the purpose of cell-contour determination using TOV-112D cells on a plastic-bottom dish.** The ability to identify cells correctly was used as a criterion for selecting a suitable transmitted-light microscopy mode. Fluorescence microscopy (fluo) of PI-stained cells was used as a reference method. Cells were manually counted in raw fluorescence images, and these numbers were used as a reference. The example image in this figure contains 583 cells. **Panel A** shows raw (red-framed) images. The fluorescence image was in-focus. The three transmitted-light images were off focus to facilitate efficient image thresholding for contrast increase. **Panel B** shows processed (green-framed) images to facilitate cell-contour identification. The determined cell contours (magenta) are overlaid with the images of the processed cells. The raw fluorescence image was converted from RGB to the 16-bit format and the background was subtracted using the "rolling ball radius" algorithm (50 pixels). The raw transmitted-light images were subjected to thresholding and converted to the 16-bit format. The percentages of correctly identified cells were: 96% in the edited fluorescence image, 88% in the edited BF image, 79% in the edited DIC image, and 43% in the edited PC image.

details on the thresholding procedure. The cells in multiple adjacent fields of view were counted in each of the four processed images with the cell-contour determination software (StarDist) using a radius range filter (3 to 12 μm) to ensure that we only counted single cells and excluded cell debris or indistinguishable clustered cells. The cell numbers obtained from the processed images were compared to the reference numbers obtained via manual counting.

Since it is known a priori that DIC is poorly suited for imaging cells on birefringent materials such as plastics, we performed a comparative study of different transmitted-light modes on TOV cells that were grown on both plastic (30-mm) and glass-bottom (50-mm) dishes. For the plastic-bottom dish, we found that the software could identify 98 ± 1%, 83 ± 5%, 68 ± 8%, and 47 ± 4% of cells in fluorescence, BF, DIC, and PC images, respectively (averaging was performed over multiple fields of view). For the glass-bottom dish, we found that the software could identify 99 ± 1%, 75 ± 7%, 70 ± 5% of cells in fluorescence, BF, and DIC images, respectively. Although the software identified 74 ± 7% single cells in PC images on a glass-bottom dish, it was clear that almost all identified cells had incorrect contours, and for this reason, PC on glass-bottom dishes was excluded from any further consideration.

The best cell-counting result was obtained for the fluorescence mode. Such a result was anticipated as fluorescence gives excellent contrast without contrast enhancement. The results for BF, DIC, and PC differ from each other beyond experimental error; however, performances of BF, DIC, and PC depend on hard-to-control experimental parameters. Therefore, instead of suggesting the blind use of BF (on either a plastic or glass-bottom dish), we recommend that users of this workflow conduct a similar experiment and determine a preferable mode for every specific experimental setting. As BF imaging of cells on a plastic-bottom dish was a winner in our competition, we adopted this mode for cell-contour identification and tracking in our work.

It is important to note that our thresholding method inevitably leads to minor loss of cell area through background removal. Since we are interested in kinetics of fluorescence intensities rather than the actual intensity values, the small and consistent loss of cell area should not influence the results significantly. Nonetheless, we demonstrated experimentally that similar rate constant distributions were obtained with different recognized cell diameters (areas) (Note S3 in S1 File). Therefore, it is appropriate to use our thresholding method for processing transmitted-light images, as the results of CRRC are unaffected by the systematic underestimation of cell areas.

### 3.3. Assumption of cell immobility during acquisition of two consecutive images

There is a short but finite time interval of a few seconds between a transmitted-light image and an accompanying fluorescence image in our new workflow. To evaluate the effects of cell movement during this short time period on the CRRC results, we performed time-lapse imaging of highly motile cells with high-frequency image acquisition for recording cell tracks (Fig 3). By using the migration tracks, we found that the speed of cell migration did not follow the normal distribution (Note S3 in S1 File). The peak of the distribution was at approximately 150 μm/h and the interquartile range was 40 μm/h. The fastest cell in the image had a speed of $v \gg 400$ μm/h. A maximum time gap between acquiring adjacent transmitted-light and fluorescence images is approximately $t_1 = 3.0$ s. The average shift of the fastest cell during this short time was $x = vt_1 = 0.33$ μm while the cell diameter was $d = 13 \pm 3$ μm. The error that such a shift in cell position can cause in the integration of intracellular fluorescence intensity over the area within cell contours is of the order of $x/d \gg 0.025$ (Note S5 in S1 File). Accordingly, the error in intracellular fluorescence intensity introduced by a finite time interval between the

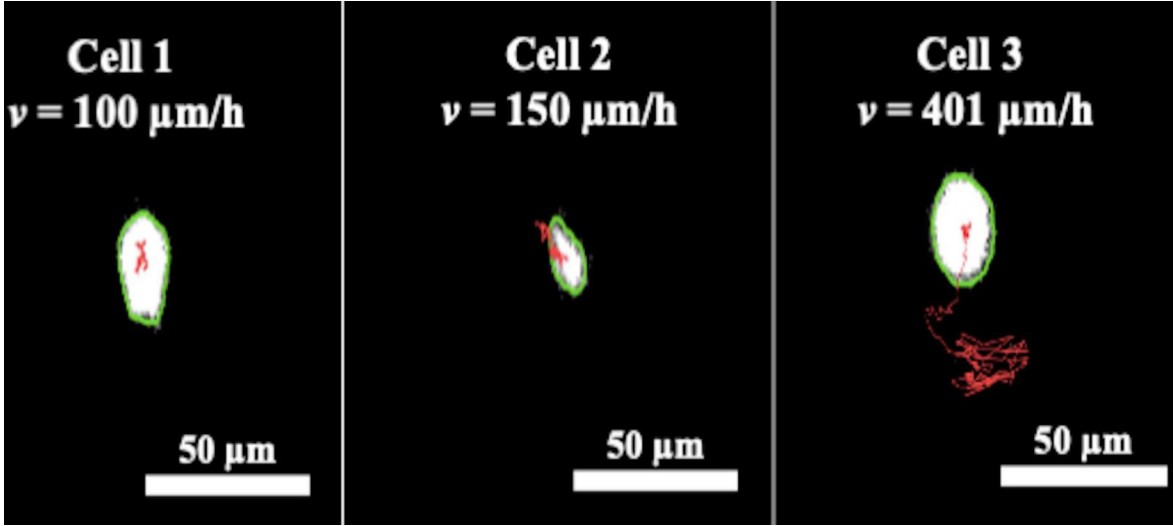

**Fig 3. Determination of speed for motile (TOV-112D) cells from cell tracks obtained with high-frequency time-lapse BF imaging (1 image per 10 s).** The three panels show representative cells with different levels of motility; red lines show respective tracks. Cell contours (green) show cell positions at the beginning of time-lapse imaging. Average speeds are shown in the panels.

transmitted-light image and an accompanying fluorescence image is approximately 2.5%, i.e., negligible, even for the fastest moving cells. Therefore, cell positions in these two images can be assumed to be identical.

We would like to re-emphasize that most advanced microscopes have options of separate focusing in both transmitted-light and fluorescence imaging modes. To demonstrate that our workflow can be also used for microscopes without such an option, we conducted a set of experiments described in Note S6 in S1 File.

### 3.4. Testing the new CRRC workflow

The original and new workflows are schematically depicted in Fig 4. To compare these two workflows and assess their sensitivity to cell motility, we performed a CRRC study of cross-membrane transport in TOV-112D cells. To favour accurate cell tracking in the new workflow, we set the time gap between adjacent transmitted-light images ($t_2$) to be shorter than the time required for the fastest cell (with speed $v$) to cover a distance equal to a typical cell diameter $d$: $t_2 << d/v$. Hence, using the values of $v = 400$ μm/h and $d = 13$ μm, we set $t_2 = 1$ min (see the previous section). Then, the two workflows were used to process the time-lapse images in parallel and obtain time dependencies (kinetic traces) of fluorescence intensities for individual cells. Kinetic traces were fitted with a single exponential decay function to find the unimolecular rate constant $k_{efflux}$ for every single cell. The kinetic traces and the results of the exponential fitting are archived in a supporting raw-data file: kinetictraces_and_fittingresults.zip.

To examine the sensitivity of both workflows to cell motility, we compared kinetic curves corresponding to cells with low and high motility. We found that the two workflows expectedly produced drastically different $k_{efflux}$ values for high-motility cells due to the inconsistency between the cell-contour mask and actual cell position (see example in Fig 5A). On the contrary, the two workflows returned similar values of $k_{efflux}$ for the low-motility cells (see example in Fig 5B); this result served as cross-validation for the two workflows. Refer to Note S7 in S1 File for a detailed analysis of the kinetic curve for the highly-motile cells shown in Fig 5A (left-panel).

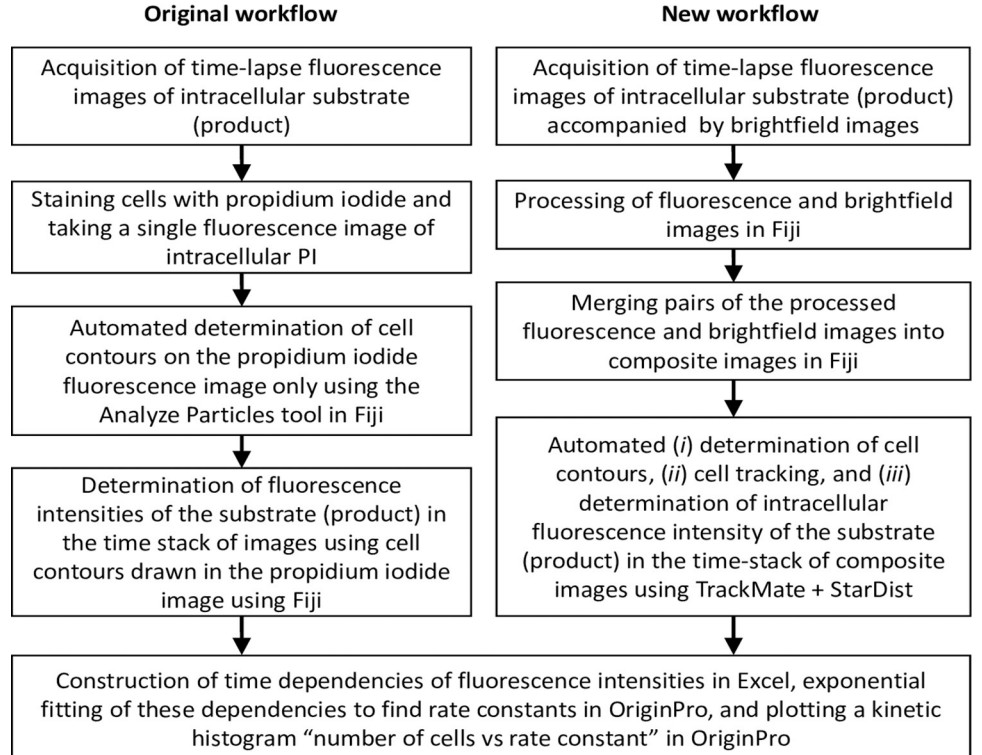

**Fig 4.** Schematic depictions of the original (left) and new (right) workflows. The last step is identical for both workflows.

An important conclusion from the detailed comparison of fluorescence-decay kinetics of cells with different motility is that the original workflow tends to overestimate the rate constant of substrate efflux for high-motility cells. This necessarily leads to the shift of the CRRC histogram produced by the original workflow to the right when compared to the histogram obtained with the new workflow (Fig 6). Importantly, a similar overestimation of $k_{\text{efflux}}$ values is observed with both OriginPro and a custom-made fitting program. Another important observation is that the overestimation of rate constant in the original workflow can falsely identify a subpopulation of cells with high rate constants.

We used a non-parametric statistical test to examine whether there was a significant difference in the kinetic constant ($k_{\text{efflux}}$) distributions produced by the two workflows. The Kolmogorov-Smirnov test confirmed that the histograms in Fig 6 differed significantly at the 0.001 significance level ($D = 0.376$, $D_\alpha = 0.209$, $p = 2.82 \times 10^{-11}$ (see Note S8 in S1 File for details on our statistical analysis). Note, the two distributions in Fig 6 have different sample sizes; this occurs since the two workflows differ in their cell-segmentation steps. The Kolmogorov-Smirnov test is insensitive to differences in sample size. Therefore, based on these results we can conclude that the new workflow produces a different and more accurate histogram due to its insensitivity to cell motility.

## 4. Conclusions

We reported on the development of a new CRRC workflow which features automated cell identification and cell tracking in transmitted-light microscopy. Such a workflow can be used for analysing a wide scope of cell types and can be considered an important move towards

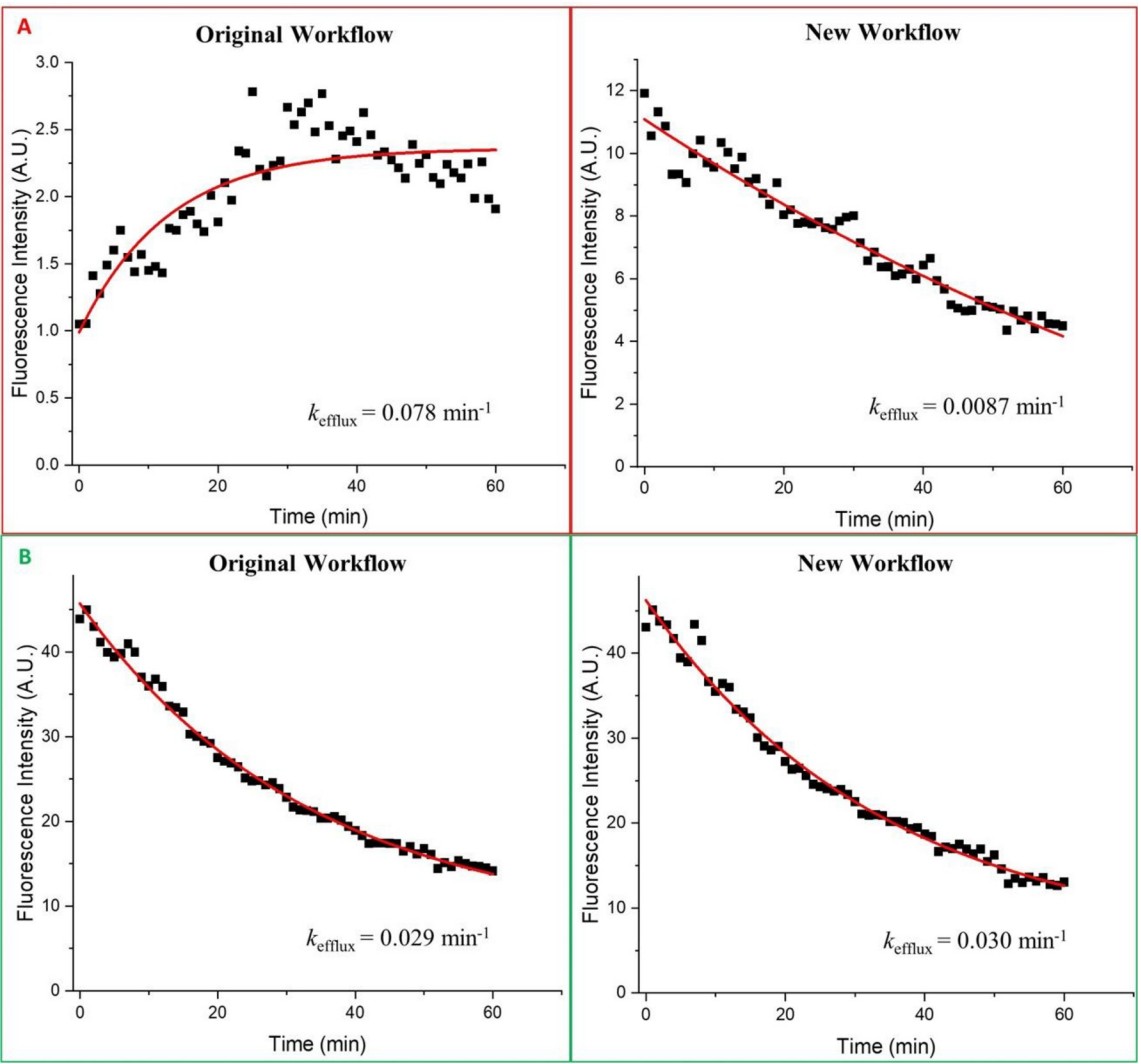

**Fig 5. Examples of kinetic curves obtained using the original and new workflows.** The data from the four different curves was fitted to the exponential decay function in OriginPro (ExpDec1 function) and a custom-made curve-fitting program. The line-of-best-fit is shown in red. **(A)** High-motility cell. The original workflow produces a curve that is not a single exponential decay. Both curve-fitting programs do not reject the curve giving a $k_{efflux}$ value which is 9-fold greater than the one obtained from the new-workflow curve. **(B)** Low-motility cell. The two workflows compute almost identical kinetic curves and $k_{efflux}$ values.

making CRRC a practical analytical tool for cytometry studies. Our new workflow will allow researchers to start CRRC studies of a wide range of intracellular enzymatic reactions in different types of cells, including highly motile cells. In recent years, there has been significant progress in rational design of high-quality fluorogenic substrates for intracellular enzymes. Specifically, such substrates have been created for enzymes responsible for chemoresistance of cancer tissues: aldehyde dehydrogenase [34, 35], and cytochrome P450 [36]. We foresee that combining our new CRRC workflow with these substrates will help discover and validate new types of predictive biomarkers of chemoresistance [13]. Finally, the cell-segmentation/cell-tracking tool disclosed here represents an entry-level, user-friendly option that can be used for a variety of biological assays (i.e., migration, proliferation, etc.) and requires no prior knowledge of informatics (i.e., training a model for deep learning).

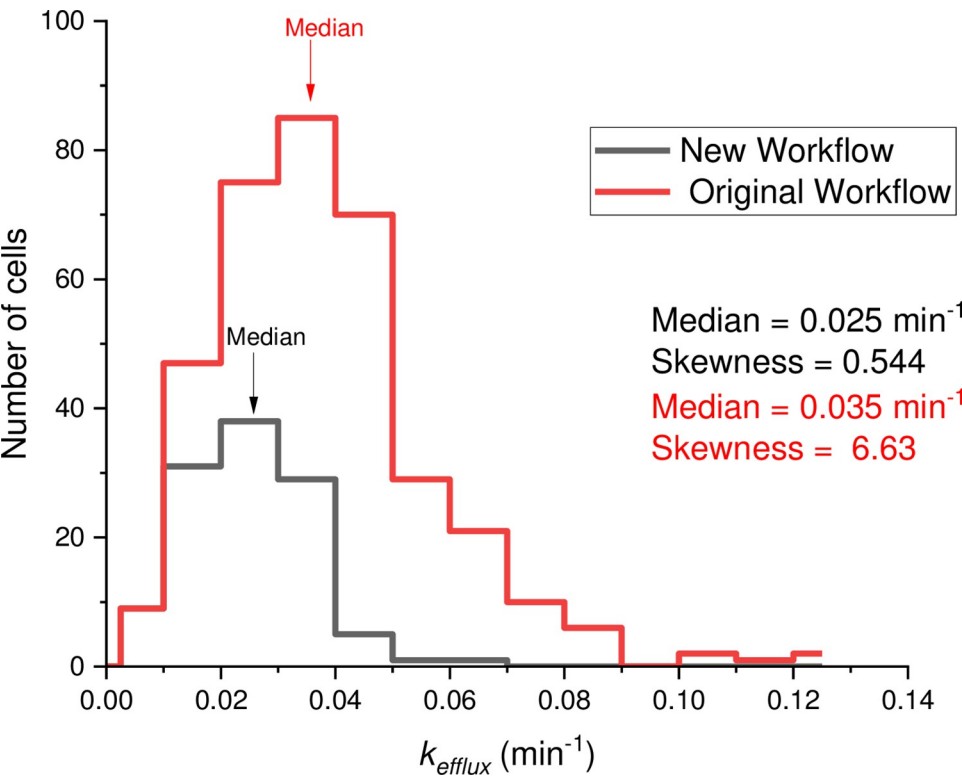

**Fig 6. CRRC final histograms of cross-membrane transport activity in TOV-112D cells.** The variation in sample size is due to differences in cell-segmentation and filtering processes. Both, median and skewness values are shown; the location of the median values on the graph are indicated with arrows. The histogram obtained from the original workflow is clearly skewed towards the right. The two distributions were found to be statistically different by the Kolmogorov-Smirnov test at the 0.001 significance level ($p = 2.82 \times 10^{-11}$).

## Supporting information

**S1 File. This is our supporting information file which contains a comprehensive list of additional data as cited in the main text.**
(PDF)

## Author Contributions

**Conceptualization:** Giammarco Nebbioso, Robel Yosief, Vasilij Koshkin, Vadim Elisseev.

**Data curation:** Giammarco Nebbioso, Robel Yosief.

**Formal analysis:** Giammarco Nebbioso.

**Investigation:** Yumin Qiu.

**Resources:** Yumin Qiu, Chun Peng.

**Supervision:** Chun Peng, Sergey N. Krylov.

**Validation:** Giammarco Nebbioso.

**Visualization:** Giammarco Nebbioso.

**Writing – original draft:** Giammarco Nebbioso, Robel Yosief, Sergey N. Krylov.

**Writing – review & editing:** Giammarco Nebbioso, Robel Yosief, Sergey N. Krylov.

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
