## [Decision Letter · Decision Letter 0]

20 Jan 2023

PONE-D-22-31376Workflow for Cytometry of Reaction Rate Constant (CRRC) to Be Applicable to Motile CellsPLOS ONE

Dear Dr. Krylov,

Thank you for submitting your manuscript to PLOS ONE. After careful consideration, we feel that it has merit but does not fully meet PLOS ONE’s publication criteria as it currently stands. Therefore, we invite you to submit a revised version of the manuscript that addresses the points raised during the review process.

We look forward to receiving your revised manuscript.

Kind regards,

Xiaojun Ren, Ph.D.

Academic Editor

PLOS ONE

Journal Requirements:

    "This work was supported by the Natural Sciences and Engineering Research Council of Canada (grant STPG-P 521331-2018 to SNK) and the Canadian Institutes of Health Research (grant PJT-166079 to CP)."

   "This work was supported by the Natural Sciences and Engineering Research Council of Canada https://www.nserc-crsng.gc.ca/index_eng.asp (grant STPG-P 521331-2018 to SKN; Sergey N. Krylov) and the Canadian Institutes of Health Research https://cihr-irsc.gc.ca/e/193.html (grant PJT-166079 to CP; Chun Peng).

Reviewers' comments:

Reviewer's Responses to Questions

**Comments to the Author**

1. Is the manuscript technically sound, and do the data support the conclusions?

Reviewer #1: Yes

2. Has the statistical analysis been performed appropriately and rigorously? 

Reviewer #1: Yes

3. Have the authors made all data underlying the findings in their manuscript fully available?

Reviewer #1: Yes

4. Is the manuscript presented in an intelligible fashion and written in standard English?

Reviewer #1: No

5. Review Comments to the Author

Reviewer #1: This manuscript addresses the flaws of existing methods for measuring CRRC quite well. However, few issues exist which need to be considered and corrected. The new CRPC workflow needs to be discussed extensively with citing relevant literature which is mainly lacking. Title should be revised and can be more scientific than technical. No references have been cited in second part of introduction section line 17-36. All figure captions should be listed at the end and should not be scattered in the manuscript. English should be revised.

6. PLOS authors have the option to publish the peer review history of their article (what does this mean?). If published, this will include your full peer review and any attached files.

Reviewer #1: No

---

## [Author Response · Author response to Decision Letter 0]

29 Jan 2023

COMMENTS TO THE AUTHOR

REVIEWER #1

Comment 1: The new CRPC workflow needs to be discussed extensively with citing relevant literature which is mainly lacking. No references have been cited in second part of introduction section line 17-36. 

Response and changes made: We have added a section to the introduction to place our new workflow in the context of current literature. References 14-23 have been added to the second part of the introduction.

Comment 2: Title should be revised and can be more scientific than technical.

Response and changes made: We replace the original title with the following: “Automated identification and tracking of cells in cytometry of reaction rate constant (CRRC)”

Comment 3: All figure captions should be listed at the end and should not be scattered in the manuscript. English should be revised.”

Response and changes made: all figure captions are now listed at the end of the paragraph where they are first cited.

Comment 4: English should be revised.

Response: The manuscript was carefully proofread for English. Changes made are highlighted ion the “changes tracked” version of the revised manuscript.

---

## [Editor Report · Decision Letter 1]

1 Mar 2023

Automated identification and tracking of cells in cytometry of reaction rate constant (CRRC)

PONE-D-22-31376R1

Dear Dr. Krylov,

We’re pleased to inform you that your manuscript has been judged scientifically suitable for publication and will be formally accepted for publication once it meets all outstanding technical requirements.

Kind regards,

Xiaojun Ren, Ph.D.

Academic Editor

PLOS ONE
---

## [Editor Report · Acceptance letter]

3 Mar 2023

PONE-D-22-31376R1 

Automated identification and tracking of cells in cytometry of reaction rate constant (CRRC) 

Dear Dr. Krylov:

I'm pleased to inform you that your manuscript has been deemed suitable for publication in PLOS ONE. Congratulations! Your manuscript is now with our production department. 

Kind regards, 

on behalf of

Dr. Xiaojun Ren 

Academic Editor

PLOS ONE